# Effect of the HXBM408 bacteria on rat intestinal bacterial diversity and the metabolism of soybean isoflavones

**Li Xiaobin[⊙], Xie Jinglong[⊙], Zhao Fang[‡], Wang Chenchen[‡], Yang Kailun[ID]***

Xinjiang Key Laboratory of Meat & Milk Production Herbivore Nutrition, Xinjiang Agricultural University, Urumqi, China

⊙ These authors contributed equally to this work.
‡ ZF and WC also contributed equally to this work.
* 40143084@qq.com

**Data Availability Statement:** All relevant data are within the paper and its Supporting Information files.

## Abstract

The purpose of this study was to investigate the effect of the HXBM408 bacteria on the diversity of rat intestinal bacteria and the metabolism of soybean isoflavones. The control group was administered sterilized water and daidzein by gavage for 7 days. Conversely, the experimental group was administered HXBM408 solution and daidzein by gavage for 7 days. The content of the daidzein metabolite equol in rat feces in the experimental group was significantly higher than that in the control group ($P < 0.05$) on the 7th and 14th days. However, the content of daidzein and its metabolites in feces was not significantly different ($P > 0.05$). On the 7th day, the relative abundance of *Streptococcus* in the feces of the experimental group was significantly higher than that of the control group ($P < 0.05$), but the difference disappeared over time ($P > 0.05$). In the intestinal digesta of rats, the proteobacteria of the experimental group was significantly lower than those of the control group ($P < 0.05$). HXBM408 can increase the degradation ability of soybean isoflavones in a short period after ingestion, increase the number of beneficial intestinal flora, and improve the structure of the flora.

## 1 Introduction

Soy isoflavones are polyphenolic compounds that have biological activities such as anti-cancer [1], menopausal syndrome relief [2], and cardiovascular disease protection [3]. After ingestion of soy isoflavones, animals can metabolize dihydrodaidzein (DHD), equol (Eq), and O-desmethylangolensin (O-Dma) under the action of specific bacteria in the gastrointestinal tract [4], these metabolites have higher biological activity than their parent compounds [5, 6]. The degradation of isoflavones by individuals occurs in two ways: one is the presence of soy isoflavone-degrading bacteria in the animal's gastrointestinal tract, and the other is because of the presence of soy isoflavones in the diets consumed by animals. Due to the differences in the structure of the gut microbiota among individuals, not all individuals have bacteria that can degrade soy isoflavones [7, 8]. Understanding how soy isoflavone-degrading bacteria survive

**Funding:** This work was supported by the Ministry of Science and Technology of Xinjiang Uygur Autonomous Region, Autonomous Region Key Laboratory Project (grant number 2016D03012).

**Competing interests:** The authors have declared that no competing interests exist.

in the gastrointestinal tract of animals will make it possible to increase the ability of animals to degrade soy isoflavones. Xun Z et al [9] found that there was no significant difference in the content of EQ in the feces of rats fed the mixture of Eq-producing strains ZX7 and daidzein (DAI) compared with the rats fed Eq directly, indicating that the rat's ability to degrade DAI may be changed after oral administration of Eq-producing bacteria. However, since the rat itself has a strain capable of degrading DAI, the degradation of DAI is likely due to the role of intragastric or intestinal bacteria.

The isolated bacteria that have the ability to degrade soy isoflavones are less used in animals, and are only used to research animal metabolism of soybean isoflavones. However, after the exogenous strain enters the animal's intestine, it may change the original bacterial flora structure in the host's intestine. Qin H et al [10] supplemented fattening sows with *Lactobacillus acidificans* and significantly increased the number of lactic acid bacteria in the jejunum. The number of *E. coli* was significantly reduced, which changed the intrinsic bacterial structure of the sow itself and had a beneficial effect on the host. However, Guang Y et al [11] found that exogenous *Helicobacter pylori* (Hp) can interact with microorganisms in the gastrointestinal tract, causing a significant increase in the number of Lactobacilli, Actinomyces, and Bacteroidetes in the gastrointestinal tract. The number of thick-walled bacteria was significantly reduced, leading to a detrimental effect on the host. After the isoflavone-degrading bacteria are used as exogenous bacteria to enter the animal's intestine, they have increased the ability of the animals to degrade soy isoflavones, and at the same time, they have no effect on the endogenous bacterial flora. Therefore, in this study, the soybean isoflavone transformant HXBM408 (*Pediococcus acidilactici*) isolated from the fresh feces of pregnant horses was cultured in vitro, and the cultured fresh broth was orally administered to SD rats to investigate the transformation of soybean isoflavones.

## 2 Materials and methods

### 2.1 Experimental animals

Twenty female Sprague-Dawley rats with an average body weight of $160.00 \pm 10.00$ g and good health status (Xinjiang Medical College) were selected.

### 2.2 Chemicals and instruments

Brain–heart infusion (BHI) was from Hopebip (Qingdao). Azithromycin (in BHI) was from SHIYAO Group OUYI Pharmaceutical Co. Ltd. (Shijiazhuang). β-Glucuronidase/Sulfatase was from Helix pomatia. Daidzein and dihydrodaidzein were from Sigma (St. Louis, MO). HPLC-grade acetonitrile and methanol were from J.T. Baker (Phillipsburg, N.J.). Carbon dioxide (99.9%) and nitrogen (99.99%) were from Miquan of Urumqi.

The HPLC system was from Shimadzu (Japan). This system included binary gradient elution (LC-20AB), a UV detector (SPD-20A) and a column temperature control box (CTO-lOAS).

### 2.3 Experimental strains and preparation

**2.3.1 Experimental strains.** *Pediococcus acidilactici* HXBM408 was isolated from fresh feces of pregnant horses (GenBank database accession number: MF992210; China General Microorganism Strain Collection Center Number: 1.16375).

**2.3.2 Preparation of fresh bacteria solution.** The HXBM408 strain was inoculated in BHI broth for 14 h, after which the culture solution was centrifuged at 3500 r/min for 10 min,

**Table 1. Test group design.**

| Group | Rats | Gastric fluid | Gastric volume |
|---|---|---|---|
| Control group | 10 | sterilized and DAI | 1 mL/d |
| Trial group | 10 | HXBM408 and DAI | 1 mL/d |

the supernatant was discarded, and an appropriate amount of sterilized water was added to prepare fresh broth.

## 2.4 Experimental design

The study was approved and supported by The Animal Welfare and Ethics Committee of Xinjiang Agricultural University (Animal protocol number: 2017016).

Twenty famale SD rats were randomly divided into a control and an experimental group, with 10 rats 10 in each group (Rats purchased at Xinjiang Medical College,). All experimental rats were treated with oral azithromycin (AZI) by oral gavage at a dosage of 1 g/kg (100 times the adult dose). Each day, rats were fed at regular intervals, and the empty material and empty water were fed for 8 h, once a day, and the continuously for 1 week. After that, mice in the experimental group were orally administered with 1 mL of fresh bacterial fluid (viable cell count $3.0 \times 10^9$ CFU) and DAI (10 mg/kg) by gastric gavage strain HXBM408. The control group was sterilized with 1 mL of sterile water and DAI (10 mg/kg), fed continuously for 1 week, the test group is shown in Table 1.

## 2.5 Sample collection and determination

**2.5.1 Sample collection.**   The rats were given fresh broth for the first day and were continuously fed for 7 days. On the 7th, 14th, 21st, 28th, and 35th days of the experiment, fresh feces were collected from the mice and stored at −20˚C for testing.

On the 35th day of the experiment, the rat's blood was taken via a heparin sodium tube. The blood was centrifuged at 3000 r/min for 10 min, and the supernatant was collected and stored at −20˚C until further testing. The rats were then dissected, and the rat intestine was isolated. The contents of the intestine were collected in cryovials and stored in liquid nitrogen.

**2.5.2 Sample processing.**   *2.5.2.1 Sample handling of fecal and intestinal contents.* Feces or intestinal contents (~0.2 g) were added to 3 ml of ether for extraction. The extract was dried at 37˚C under a nitrogen blower, dissolved in 0.5 ml of methanol, and filtered through a 0.45-μm microporous membrane. The sample was kept at 4˚C for HPLC detection.

*2.5.2.2 Plasma sample processing.* Plasma (0.5 ml) was placed into a 10-ml Eppendorf tube and mixed with 3 mL of 0.1 M sodium acetate buffer pH 5.0 and 80 μl of β-glucuronidase/sulfatase (Helix pomatia). Enzymatic hydrolysis occurred in a 37˚C water bath for 12 h. Then, the tube was removed from the 37˚C water bath and placed in tap water to cool. Five ml of cyclohexane:ethyl acetate at a 1:1 (v/v) extract were added, and the solution was vortexed for 1 min and centrifuged at 5000 r/min for 10 min at 4˚C. After centrifugation, the supernatant was sparged with nitrogen to a dry state, dissolved in 200 μl of methanol, filtered through a 0.45-μm filter, and stored at 4˚C for testing [12].

**2.5.3 HPLC analysis.**   A 20 μL volume of the sample was injected and separated over a Welch Ulimate C18 column (250 × 4.6 mm, 4 μm). The temperature was set at 30˚C, and the flow rate was maintained at 0.6 ml/min. Elution was isocratic with a mobile phase consisting of methanol: acetonitrile: water (20:30:50). Daidzein and DHD were detected at 260 nm. Equol were detected at 205 nm.

**Table 2. Concentration (μg/g) of daidzein and its metabolites in rat feces.**

| Groups | | 7 d | 14 d | 21 d | 28 d | 35 d |
|---|---|---|---|---|---|---|
| DAI | Control group | 1.04±0.30[a] | 1.09±0.23 | 1.22±0.31 | 1.39±0.33 | 1.91±0.21 |
| | Trial group | 0.63±0.20[b] | 0.82±0.3 | 0.92±0.17 | 1.06±0.19 | 1.78±0.17 |
| DHD | Control group | 0.92±0.16[b] | 0.89±0.25 | 0.88±0.17 | 0.98±0.25 | 1.31±0.39 |
| | Trial group | 1.14±0.14[a] | 1.09±0.27 | 0.96±0.18 | 1.03±0.36 | 1.36±0.22 |
| Eq | Control group | 31.51±2.57[b] | 25.34±2.88[b] | 24.46±2.14 | 21.88±2.60 | 19.23±3.41 |
| | Trial group | 35.20±3.90[a] | 29.84±2.74[a] | 27.47±4.71 | 24.47±4.12 | 20.66±2.90 |

Note: In the same row, the values with different lowercase superscripts indicate a significant difference ($P < 0.05$), whereas those with same lowercase letter or no superscript letters indicate no significant difference ($P > 0.05$). In the same row, the values with a different capital letter indicate an extremely significant difference ($P < 0.01$).

**2.5.4 Analysis of biodiversity in feces and intestines.** Sequencing of bacteria in rat feces and intestinal contents was assisted by Beijing Nuohe Zhiyuan Technology Co. Ltd.

**2.5.5 Data processing.** *2.5.5.1 Data processing of daidzein and its metabolites in rat feces and plasma*. The experimental data were collated with Excel and analyzed using an independent sample *t*-test (SPSS 17.0 software). Analysis of variance was performed on the $P < 0.05$ generation. $P < 0.01$ represented a significant difference. The test results are expressed as the mean ± standard deviation.

*2.5.5.2 Microbial bacterial diversity data processing*. Metastats analysis using the R software was used to perform permutation tests between groups at each classification level (Phylum, Family, and Genus) to obtain a *p*-value, and then Benjamini and Hochberg False Discovery Rate methods were used to correct *p*-values. The q values were obtained. Species analysis with significant differences between groups was performed using R software for inter-group *t*-tests.

# 3 Results

## 3.1 Effect of intragastric administration of HXBM408 on daidzein and its metabolites in rat feces

As can be seen from Table 2, after intragastric administration of HXBM408, the DAI content in the feces of rats in the test group was 39.4% lower than that in the control group ($P < 0.05$). The contents of DHD and Eq in the test group were 23.9% and 11.7% higher, respectively, than those in the control group ($P < 0.05$). The Eq content in the feces of the test group on the 14th day was 17.8% higher than that of the control group ($P < 0.05$). On the 21st, 28th, and 35th days of the experiment, the contents of DAI, DHD, and Eq in feces did not significantly differ between the test group and the control group ($P > 0.05$).

## 3.2 Effect of intragastric administration of HXBM408 on daidzein and its metabolites in rat plasma

As shown in Table 3, after the administration of HXBM408 bacilli in rats, the contents of DAI, DHD, and Eq in plasma were not significantly different between the test group and the control group ($P > 0.05$).

**Table 3. Concentration (ng/ml) of daidzein and its metabolites in rat plasma.**

| Groups | DAI | DHD | Eq |
|---|---|---|---|
| Control group | 56.50±7.62 | 603.19±63.57 | 212.74±26.61 |
| Trial group | 56.47±7.26 | 638.79±76.58 | 212.90±29.75 |

Table 4. Alpha diversity of bacterial flora in rat feces.

| Items | | Good's(%) | Observed species | Richness estimator | | Diversity index | |
|---|---|---|---|---|---|---|---|
| Dates | Groups | | | ACE | Chao1 | Shannon | Simpson |
| 7 d (feces) | Control group | 99.85 | 282.60 | 342.72[a] | 336.08[a] | 4.06 | 0.82 |
| | Trial group | 99.89 | 250.60 | 290.80[b] | 285.76[b] | 4.49 | 0.90 |
| 21 d (feces) | Control group | 99.81 | 409.56 | 476.64 | 467.10 | 5.11 | 0.92 |
| | Trial group | 99.82 | 401.44 | 467.78 | 459.54 | 5.16 | 0.92 |
| 35 d (feces) | Control group | 99.81 | 444.20 | 505.67 | 509.97 | 5.20 | 0.90 |
| | Trial group | 99.83 | 387.38 | 451.36 | 440.55 | 4.34 | 0.81 |
| 35 d (Intestinal) | Control group | 99.83 | 509.22 | 552.52 | 545.53 | 5.64 | 0.92 |
| | Trial group | 99.83 | 473.89 | 526.64 | 513.61 | 5.39 | 0.90 |

## 3.3 Effect of intragastric administration of HXBM408 on intestinal contents and fecal bacterial flora in rats

**3.3.1 Alpha diversity of bacterial flora in rat feces and intestinal contents.** As can be seen from Table 4, the sequencing coverage of bacterial flora in rat feces was greater than 99.8%, indicating that the sequencing of this test can cover more than 99.8% of the bacterial types. The results of α-diversity index showed that on the 7th day of the test, the ACE and Chao1 of the test group were significantly lower than the control group ($P < 0.05$), but there was no significant difference between the Shannon and Simpson indices ($P > 0.05$). After HXBM408, the richness of fecal bacterial flora decreased, but there was no significant effect on the bacterial flora diversity. On the 21st and 35th days of the experiment, no significant difference was observed between the ACE, Chao1, Shannon, and Simpson control groups and experimental group ($P > 0.05$); the sequence coverage rate of bacterial flora in the intestinal contents of rats was >99.8%, indicating that the sequencing of this test can cover >99.8% of bacterial types. The results of the α diversity index showed that there was no significant difference between the ACE, Chao1, Shannon and Simpson test groups and the control group ($P > 0.05$).

**3.3.2 Beta diversity.** The results of the Principal Component Analysis (PCA) based on OTU level are shown in Fig 1. On the seventh day of the experiment, the first principal coordinate (PC1) in rat feces represented a representative contribution rate of 14.36% to the detected microbial bacteria, and the representative contribution of the second principal coordinate (PC2) to the detected microbial bacteria was 9.94%. On the 21st day of the experiment, PC1 in rat feces represented a representative contribution rate of 11.94% to the detected microbial bacteria, and the representative contribution rate of PC2 to the detected microbial bacteria was 9.7%. On the 35th day of the experiment, the representative contribution rate of PC1 in the rat feces to the detected microbial bacteria was 17.12%, and the representative contribution of PC2 to the detected microbial bacteria was 12.91%; PC1 in the intestinal contents of rats was a representative contribution rate of the detected microbial bacteria of 15.73%, and the representative contribution of PC2 to the detected microbial bacteria was 11.51%. Each group of samples was more concentrated on the 7th day, and then they could be clearly separated with the extension of time. Moreover, some of the individuals in the control group were significantly different from the experimental group, indicating that gastric gavage could change the composition of bacterial flora in rat feces.

**3.3.3 Effect of Gavage of HXBM408 bacteria on relative abundance of bacteria in rat feces and intestines.** *3.3.3.1 Effect of feeding HXBM408 bacteria on bacterial abundance in rat feces and intestinal digesta by phylum.* From Table 5, it can be seen that the bacteria in the rat

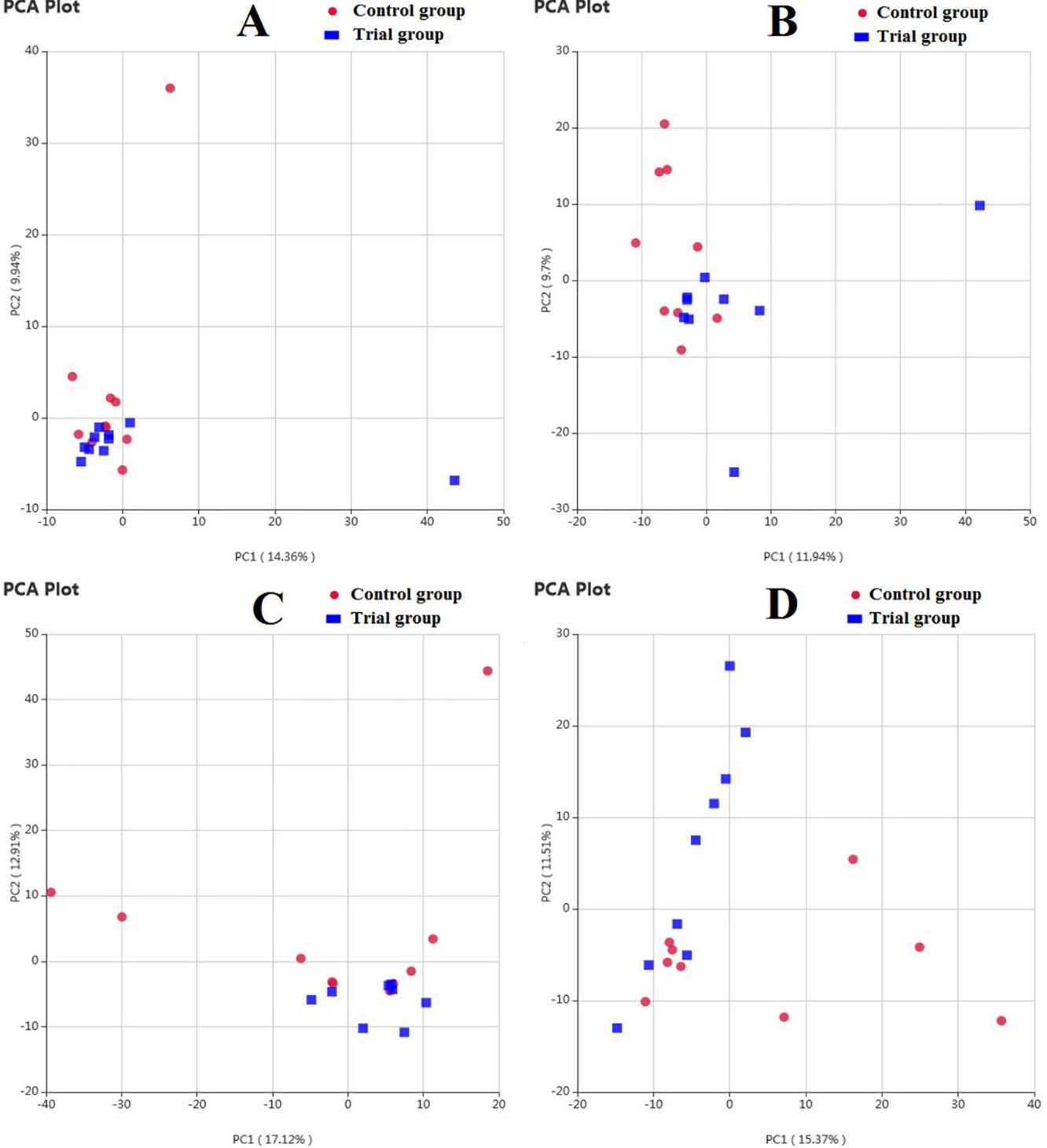

**Fig 1. The PCA profile of bacterial flora in rat feces and intestinal digesta.** A) 7 d (feces); B) 21 d (feces); C) 35 d (feces); D) 35 d (intestinal). Red: Control group; Blue: Trial group.

feces and intestinal tract can be detected at the level of the phylum Firmicutes, Bacteroidetes, Saccharibacteria, Proteobacteria, Actinobacteria, Tenericutes, Acidobacteria, Verrucomicrobia, Fusobacteria, and Peregrinibacteria. Mainly for Firmicutes and Bacteroidetes, the

**Table 5. Effect of feeding HXBM408 on the abundance of bacterium in rat feces and intestinal digesta and by phylum (%, n = 10).**

| Groups | 7 d (Feces) | | 21 d (Feces) | | 35 d (Feces) | | 35 d (Intestinal) | |
|---|---|---|---|---|---|---|---|---|
| | Control group | Trial group | Control group | Trial group | Control group | Trial group | Control group | Trial group |
| Firmicutes | 50.61 | 43.26 | 48.41 | 44.89 | 66.35 | 71.32 | 75.94 | 67.72 |
| Bacteroidetes | 48.21 | 56.41 | 50.47 | 53.62 | 30.93 | 27.02 | 21.39 | 30.98 |
| Saccharibacteria | 0.72 | 0.01 | 0.09 | 0.43 | 0.39 | 0.15 | 0.28 | 0.10 |
| Proteobacteria | 0.06 | 0.09 | 0.10 | 0.23 | 0.96 | 0.19 | 1.32[a] | 0.37[b] |
| Actinobacteria | 0.40 | 0.23 | 0.90 | 0.75 | 1.26 | 1.22 | 0.94 | 0.67 |
| Tenericutes | 0.01 | - | 0.01 | 0.06 | 0.06 | 0.09 | 0.11 | 0.12 |
| Acidobacteria | - | - | - | <0.01 | 0.01 | - | - | - |
| Verrucomicrobia | <0.01 | - | - | 0.01 | <0.01 | <0.01 | - | - |
| Fusobacteria | <0.01 | <0.01 | <0.01 | - | 0.01 | - | 0.01 | <0.01 |
| Peregrinibacteria | - | - | - | - | <0.01 | - | - | - |
| Others | 0.01 | <0.01 | 0.01 | 0.02 | 0.03 | 0.01 | 0.01 | 0.03 |

Note: In the same row, the values with different lowercase superscripts indicate significant differences ($P < 0.05$), whereas those with same lowercase or no superscript letters indicate no significant difference ($P > 0.05$). In the same row, the values with difference capital letter mean that the difference was extremely significant ($P < 0.01$), −: Not detected.

abundance of the remaining bacteria is <1%. In rat feces, there was no significant difference between the experimental group and the control group at the portal level ($P > 0.05$), and the intestinal proteobacteria in the rat intestine group was significantly higher than that in the control group ($P < 0.05$). The difference at the other phylum levels was not significant ($P > 0.05$).

*3.3.3.2 Effect of feeding HXBM408 bacteria on bacterial abundance in the rat feces and intestinal digesta by genus.* From Table 6, it can be seen that bacterial genera detected in the feces and intestinal tract of rats include *Lactobacillus*, *Clostridium*, *Bacteroides*, *Ruminococcus_1*, *Turicibacter*, *Blautia*, *Streptococcus*, *Romboutsia*, *Lachnospiraceae*, and *Prevotellaceae*. On the 7th day of the experiment, the *Streptococcus* in the feces of rats was significantly higher than that in the control group ($P < 0.05$), and there was no significant difference in the levels of other genera ($P > 0.05$). On the 21st day of the experiment, *Streptococcus* in the feces of rats in the experimental group was significantly higher than that of the control group ($P < 0.01$), and there was no significant difference in the levels of the other genera ($P > 0.05$). There was no

**Table 6. Effect of feeding HXBM408 on the abundance of bacterium in rat feces and intestinal digesta by genus (%, n = 10).**

| Groups | 7 d (Feces) | | 21 d (Feces) | | 35 d (Feces) | | 35 d (Intestinal) | |
|---|---|---|---|---|---|---|---|---|
| | Control group | Trial group | Control group | Trial group | Control group | Trial group | Control group | Trial group |
| *Lactobacillus* | 35.23 | 21.71 | 27.94 | 16.52 | 11.64 | 10.89 | 7.66 | 9.19 |
| *Clostridium* | 0.24 | 0.45 | 0.42 | 2.22 | 16.90 | 30.29 | 16.26 | 18.34 |
| *Bacteroides* | 14.16 | 22.68 | 14.4 | 14.87 | 4.43 | 8.50 | 2.30[B] | 7.50[A] |
| *Ruminococcus_1* | 0.29 | 0.34 | 1.30 | 4.02 | 1.72 | 0.90 | 1.47 | 1.40 |
| *Turicibacter* | 0.01 | 0.01 | 0.03 | 0.21 | 5.48 | 7.53 | 5.74 | 4.35 |
| *Blautia* | 4.95 | 7.92 | 2.24 | 2.29 | 0.70 | 2.72 | 0.54[b] | 2.50[a] |
| *Streptococcus* | 0.01[b] | 0.36[a] | 0.01[B] | 1.62[A] | 5.82 | 1.36 | 7.42[a] | 0.53[b] |
| *Romboutsia* | 0.02 | 0.01 | 0.06 | 0.04 | 3.94 | 4.95 | 4.66[b] | 1.91[a] |
| *Lachnospiraceae* | 0.21 | 1.16 | 1.16 | 1.56 | 1.77 | 0.98 | 5.63 | 2.48 |
| *Prevotellaceae* | 0.71 | 0.01 | 5.17 | 4.41 | 3.31 | 1.48 | 0.70 | 0.51 |
| Others | 44.19 | 45.36 | 47.18 | 52.24 | 44.29 | 30.41 | 47.60 | 51.2 |

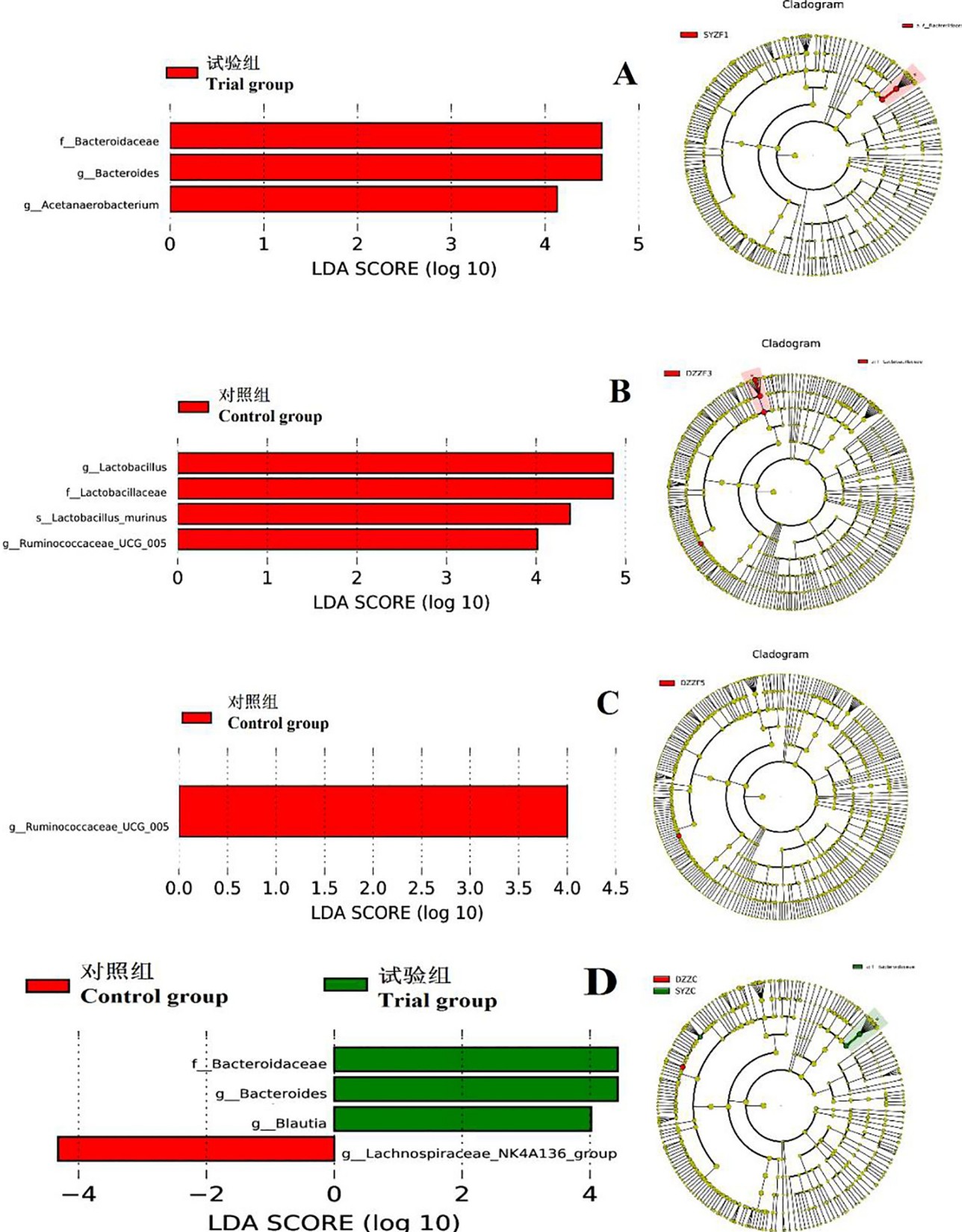

**Fig 2. The LEfSe profile of bacterial flora in rat feces and intestinal digesta.** A, 7 d (feces); B, 21 d (feces); C, 35 d (feces); D, 35 d (intestinal). Red: Control group; Green: Trial group.

significant difference in the level of each genera of the feces of the rats on the 35th day of the experiment ($P > 0.05$). The rats in the Intestinal *Blautia* group were significantly higher than the control group ($P < 0.05$). The *Bacteroides* test group was significantly higher than the control group ($P < 0.01$), whereas the *Streptococcus* and *Romboutsia* were significantly lower in the test group ($P < 0.05$). There was no significant difference in other genera levels ($P > 0.05$).

**3.3.4 LEfSe in feces and intestinal contents in rats.** LEfSe analysis results in rat feces and intestinal contents are shown in Fig 2. Panel A shows histograms and evolutionary branching plots of LDA values in the feces of rats on the 7th day of the experiment. LAD SCORE shows that the predominant flora at the classification level of the experimental group is the Bacteroidaceae, *Bacteroidetes* and *Acetanaerobacterium*. Panel B shows the LEfSe analysis of bacterial flora in the feces of rats on the 21st day of the experiment, LAD SCORE shows that the dominant predominant bacteria in the control group are Lachnospiraceae, *Lactobacillus* and *Ruminococcaceae_UGG_005*. Panel C shows the LEfSe analysis of bacterial flora in the feces of rats on the 35th day of the experiment. The LAD SCORE shows that the dominant flora in the control group is *Ruminococcaceae_UGG_005*. Panel D shows the bacteria in the intestinal fecal contents of rats. The LAD SCORE showed that the predominant bacterial group at the classification level of the control group was Lachnospiraceae_NK4A136_group, and the dominant bacterial population at the classification level of the experimental group was Bacteroidaceae, *Bacteroides*, and *Blautia*.

# 4 Discussion

Recent studies have shown that the biological activity of SIF with DAI as the main active ingredient is mainly attributed to the metabolites DHD and Eq that are degraded after being taken into the body, and the degradation process requires specific intestinal bacteria [13, 14]. Whether exogenously added DAI-degrading bacteria improve an animal's ability to degrade DAI and the effect on the overall structure of the intestinal microflora are less studied.

## 4.1 Effects of intragastric administration of HXBM408 on daidzein and metabolites in rat feces and plasma

Bowey [15] colonized GF rats with Eq-producing human feces and detected the production of Eq in the feces of the rats, indicating that exogenous strains were used to alter the ability of animals to degrade DAI. In our experiments, SD rats were gavaged with the soy isoflavone-transforming bacteria HXBM408 isolated from fresh feces of pregnant horses. The results showed that the DHD and Eq content in the feces of the test group was significantly higher than that in the control group on the 7th day of the experiment. The DAI content was significantly lower in the test group than in the control group, indicating that SD rats have improved ability to degrade DAI after intragastric administration of HXBM408. Studies have found that when the concentration of DHD is increased, bacteria that can convert DHD to Eq can grow rapidly, thereby increasing the concentration of Eq [16]. In this experiment, strain HXBM408 can only degrade DAI to produce DHD, whereas SD rat can tolerate bacteria. The increase of Eq concentration in feces after fluid may be due to the fact that the increase of DHD concentration can promote the rapid growth of bacteria that can convert DHD into Eq in rats, and then increase the Eq content in rat feces.

In this study, continuous intragastric administration of HXBM408 was stopped after one week. As time progressed, there was no significant difference in the contents of DAI and metabolites in the feces of rats in the control group and the experimental group nor in the DAI levels and its metabolism in rat plasma. There was no significant difference in the content of the substances. The reasons for these results may be as follows. First, the optimum pH value of the HXBM408 strain in the culture process was approximately 7.2, whereas the pH value of the gastric juice solution in the rat could reach 1.2, potentially resulting in the death of the strain. Afterwards, the rat' ability to degrade DAI was restored to the pre-gastric condition, as no fresh gavage was continued [17]. Secondly, the strain HXBM408 was selected from the feces of pregnant horses. Horses are herbivores, whereas rodents are omnivorous. Unused dietary structures caused an environment that was not used in the intestines of the animals. The strain HXBM408 entered the intestinal tract of the rats. It belongs to the passage flora and cannot be colonized in the intestine of rats [18, 19].

## 4.2 Effect of gavage of HXBM408 bacteria on intestinal tract contents and bacterial microflora of feces in rats

The HXBM408 strains fed with SD rats in this test belonged to the genus *Pediococcus* of the Streptococcus family. However, at the genus level, the *Pediococcus* was not detected. Conversely, the failure to detect *Pediococcus* was probably due to the death of strain HXBM408 due to the low pH of the rat's gastric juice, or due to the existence of different pathogenic bacteria between horses and rats. Therefore, it is recommended that exogenous strains first be processed by coating in future studies. The strains can reach the intestine through the stomach, or the strains can be transported to the animal's intestine through other channels, and the strains can grow and multiply in the intestine and colonize in the intestine. Intestinal species differences between different animals should also be considered.

After the rats were treated with antibiotics, their intestinal flora diversity was significantly reduced [20, 21]. Afterwards, as time increases, the microbial flora can gradually recover after the antibiotics are metabolized and excreted. After the rats were treated with antibiotics, the diversity of intestinal bacterial flora increased with time. On the 7th day of the experiment, the estimated bacterial community abundance in the experimental group was significantly lower than that in the control group. It may be possible to exogenously add strains to reduce the abundance of bacterial flora in rat feces. The results of PCA showed that as time progressed, the two groups became more and more dispersed, indicating that the exogenous broth can change the composition of bacterial colonies in the feces of rats.

The intestinal microorganisms of animals belong to the genera Streptomyces, Bacteroidetes, Proteobacteria, Actinomycetes, Acidobacilli, and Microbacterium, among which the Streptomyces and Bacteroidetes are the main dominant phyla [22, 23]. The results of this experiment showed that the rat feces and intestinal microflora at the gate level were mainly thick-walled bacteria and Bacteroidetes, and the results were consistent with Eckburg et al. Bacteroidetes play an important role in the metabolism of large DAI, and the number of Bacteroidetes is positively correlated with the content of Eq in rat feces. On the 7th day of the experiment, the Eq content of rat feces was significantly higher in the test group than in the control group, and the LEfSe analysis results showed that the dominant predominant bacterial groups in the test group were Bacteroidaceae and *Bacteroides*. The amount of Eq in feces was positively correlated with Bacteroides numbers. The level of microbial bacteria in the intestinal tract of rats was significantly lower in the test group than that in the control group, indicating that the administration of HXBM408 can inhibit the growth of harmful bacteria in the intestine of rats. It has also been found that some bacteria of the genus Eubacterium [24], Lactobacillus [25],

Butyricum [26] and Clostridium [27] in the intestine of animals are directly involved in the degradation of daidzein. Bolca [28] studied DAI-degrading bacteria in humans and found that the number of *Clostridium* in the intestinal tract and the DAI metabolite Eq content showed a significant negative correlation. We found that the Eq content of feces in the experimental rats was higher than the control group, but the *Clostridium* abundance test group was also higher than the control group. The number of *Clostridium* species was positively correlated with the Eq content. The results were contrary to the results of the Bolca study. There may be species specific differences. In addition, the levels of *Bacteroides* and *Lauterella* were significantly higher in the intestine experimental rats than in control rats, but those of the *Streptococcus* genera were significantly reduced, indicating that exogenously administered HXBM408 bacilli may regulate the structure of endogenous bacterial flora in the intestine of rats.

## Author Contributions

**Conceptualization:** Li Xiaobin, Xie Jinglong, Yang Kailun.

**Data curation:** Li Xiaobin, Xie Jinglong, Zhao Fang.

**Formal analysis:** Li Xiaobin, Zhao Fang, Wang Chenchen.

**Funding acquisition:** Yang Kailun.

**Investigation:** Xie Jinglong.

**Methodology:** Li Xiaobin, Wang Chenchen.

**Project administration:** Xie Jinglong.

**Resources:** Yang Kailun.

**Writing – original draft:** Li Xiaobin, Xie Jinglong.

**Writing – review & editing:** Li Xiaobin, Xie Jinglong, Yang Kailun.

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
