## [Decision Letter · Decision Letter 0]

10 Dec 2020

PONE-D-20-06300

Effect of the HXBM408 bacteria on rat intestinal bacterial diversity and the metabolism of soybean isoflavones

PLOS ONE

Dear Dr. Kailun,

Thank you for submitting your manuscript to PLOS ONE. After careful consideration, we feel that it has merit but does not fully meet PLOS ONE’s publication criteria as it currently stands. Therefore, we invite you to submit a revised version of the manuscript that addresses the points raised during the review process.

The revisions suggested by reviewers pertain mostly to the quality of writing and presentation of the information, including making sure the authors properly define all acronyms and making sure their figures are numbered and labelled correctly.  There are also a number of comments asking for more detail on the methods, and for the authors to reflect on certain aspects.  While minor, I do recommend that the authors take their time to consider all reviewer comments fully.  Thank you for your patience during this unexpectedly long review process.

We look forward to receiving your revised manuscript.

Kind regards,

Suzanne L. Ishaq, PhD

Academic Editor

PLOS ONE

Journal Requirements:

2. In your Methods section, please state the volume of the blood samples collected for use in your study.

3. To comply with PLOS ONE submissions requirements, in your Methods section, please provide additional information on the animal research and ensure you have included details on (1) methods of sacrifice, (2) methods of anesthesia and/or analgesia, and (3) efforts to alleviate suffering.

4. In your Ethics Statement, please confirm that this animal study is approved by IACUC in Xinjiang Agricultural University.

5. We suggest you thoroughly copyedit your manuscript for language usage, spelling, and grammar. If you do not know anyone who can help you do this, you may wish to consider employing a professional scientific editing service.  

"This work was supported by the Ministry of Science and Technology of Xinjiang Uygur

Autonomous Region, Autonomous Region Key Laboratory Project (grant number 2016D03012)."

" The author(s) received no specific funding for this work."

7. Please include your tables as part of your main manuscript and remove the individual files. Please note that supplementary tables should be uploaded as separate "supporting information" files.

Reviewers' comments:

Reviewer's Responses to Questions

**Comments to the Author**

1. Is the manuscript technically sound, and do the data support the conclusions?

Reviewer #1: Yes

Reviewer #2: Partly

2. Has the statistical analysis been performed appropriately and rigorously? 

Reviewer #1: Yes

Reviewer #2: Yes

3. Have the authors made all data underlying the findings in their manuscript fully available?

Reviewer #1: Yes

Reviewer #2: Yes

4. Is the manuscript presented in an intelligible fashion and written in standard English?

Reviewer #1: Yes

Reviewer #2: No

5. Review Comments to the Author

Reviewer #1: The authors investigate the effect of isoflavone-transforming bacteria HXBM408 isolated from fresh feces of pregnant horses on rat intestinal bacterial diversity and the metabolism of soybean isoflavones. The authors culture the soybean isoflavone transformant HXBM408 in vitro and orally administrate to SD rats to investigate the transformation of soybean isoflavones in vivo, and find that the contents of DAI, DHD and Eq is significantly different between the test group and the control group in the first 14 day post gavage of HXBM408, but the differences are not observed in plasma; in addition, they also find that the administration of HXBM408 can also have an effect on the relative abundance of bacteria in rat feces and intestines, which are quite intriguing. I would support accepting this paper to PLOS ONE after suitable revisions are made.

1. There are several abbreviations in the manuscript that have no full name annotation, like “OTU” in page 9 and “LEfSe” in page 12, the authors should check the manuscript and make sure that there is no obscure abbreviation.

2. There is no figure 1 in this manuscript and the text jumps into figure 2? (the first mentioned figure throughout this manuscript is in page 9, but the figure number is figure 2-1)

3. Several “LAD SCORE” in page 12 should be “LDA SCORE”.

4. Figure 2-2 A, B and C, page 12 of the text, the authors show the LDA SCORE of test group on 7th day, LDA SCORE of control group on 21th day and 35th. How about the LDA SCORE of control group on 7th day? And LDA SCORE of test group on 21th day and 35th?

5. Figure 2-2, there are annotation like “Control group” and “Trial group” in the figure, will it be better to remove the corresponding Chinese annotation?

Reviewer #2: The submission reported the effect of HXBM408 on the bioconversion of DAI and so as the microflora in the rat GI. Following concerns should be addressed before further consideration.

1. Why did the authors choose female Sprague-Dawley rats instead of males? You were only evaluating microflora but not female related cancers?

2. What's the age of your rats?

3. Please mentioned Pediococcus acidilactici instead of HXBM408 in you content which was misleading.

4. what's 3500 r/min? Its not the standard expression.

5. Why not the authors preform the test in a dose-depentdant manner? How did you confirm the dosage at 1 g/kg?

6. 1.5.1:

7. What's the difference b/w administration of DAI for bioconversion and direct feeding of Eq?

8. From 3.1: The results seemed not really successful, what's the comment from the authors?

9. Figure 2: Please remove the Chinese words from the content.

10. It seems that there was no long-trem effect from the feedding of HXBM408. What's the comment from the authors?

11. The English style should be revised.

6. PLOS authors have the option to publish the peer review history of their article (what does this mean?). If published, this will include your full peer review and any attached files.

Reviewer #1: No

Reviewer #2: No

---

## [Author Response · Author response to Decision Letter 0]

14 Mar 2021

1.There are several abbreviations in the manuscript that have no full name annotation, like “OTU” in page 9 and “LEfSe” in page 12, the authors should check the manuscript and make sure that there is no obscure abbreviation.

OTU/LEfSe has been added

2.There is no figure 1 in this manuscript and the text jumps into figure 2? (the first mentioned figure throughout this manuscript is in page 9, but the figure number is figure 2-1)

Picture serial number has been corrected

3.Several “LAD SCORE” in page 12 should be “LDA SCORE”.

already edited

4.Figure 2-2 A, B and C, page 12 of the text, the authors show the LDA SCORE of test group on 7th day, LDA SCORE of control group on 21th day and 35th. How about the LDA SCORE of control group on 7th day? And LDA SCORE of test group on 21th day and 35th?

Explanation has been added

5.Figure 2-2, there are annotation like “Control group” and “Trial group” in the figure, will it be better to remove the corresponding Chinese annotation?

Chinese deleted

Reviewer #2: The submission reported the effect of HXBM408 on the bioconversion of DAI and so as the microflora in the rat GI. Following concerns should be addressed before further consideration.

1.Why did the authors choose female Sprague-Dawley rats instead of males? You were only evaluating microflora but not female related cancers?

In this article, the effects of DAI and EQ are estrogen-like effects, which are mainly reflected in female animals.

2.What's the age of your rats?

Rats are 50 days old

3.3.Please mentioned Pediococcus acidilactici instead of HXBM408 in you content which was misleading.

already edited

4.what's 3500 r/min?（转速） Its not the standard expression.

already edited

5.Why not the authors preform the test in a dose-depentdant manner? How did you confirm the dosage at 1 g/kg?

According to the calculation of the bacterial viability of the bacterial liquid, each rat is fed no less than 109 live bacteria per day

6.What's the difference b/w administration of DAI for bioconversion and direct feeding of Eq?

DAI is the precursor of EQ. The price of EQ is very high. The purpose of this experiment is to make rats have the ability to convert DAI into EQ.

7.From 3.1: The results seemed not really successful, what's the comment from the authors?

After the bacteria liquid stopped gavage, the EQ transformation ability of rats did not increase over time. It may be because Pediococcus acidilactici, as a passing bacteria, failed to permanently colonize the rats, so further research is needed in the later period.

9. Figure 2: Please remove the Chinese words from the content.

Chinese deleted

10.It seems that there was no long-trem effect from the feedding of HXBM408. What's the comment from the authors?

After the bacteria liquid stopped gavage, the EQ transformation ability of rats did not increase over time. It may be because Pediococcus acidilactici, as a passing bacteria, failed to permanently colonize the rats, so further research is needed in the later period.

11. The English style should be revised.

Language has been modified

---

## [Decision Letter · Decision Letter 1]

19 Apr 2021

PONE-D-20-06300R1

Effect of the Pediococcus acidilactici bacteria on rat intestinal bacterial diversity and the metabolism of soybean isoflavones

PLOS ONE

Dear Dr. Kailun,

Thank you for submitting your manuscript to PLOS ONE. After careful consideration, we feel that it has merit but does not fully meet PLOS ONE’s publication criteria as it currently stands. Therefore, we invite you to submit a revised version of the manuscript that addresses the points raised during the review process.

The authors have done well to address the comments from the first review, but some additional detail in the methods has been requested.

We look forward to receiving your revised manuscript.

Kind regards,

Suzanne L. Ishaq, PhD

Academic Editor

PLOS ONE

Journal Requirements:

Reviewers' comments:

Reviewer's Responses to Questions

**Comments to the Author**

1. If the authors have adequately addressed your comments raised in a previous round of review and you feel that this manuscript is now acceptable for publication, you may indicate that here to bypass the “Comments to the Author” section, enter your conflict of interest statement in the “Confidential to Editor” section, and submit your "Accept" recommendation.

Reviewer #1: All comments have been addressed

Reviewer #2: All comments have been addressed

2. Is the manuscript technically sound, and do the data support the conclusions?

Reviewer #1: Yes

Reviewer #2: Yes

3. Has the statistical analysis been performed appropriately and rigorously? 

Reviewer #1: Yes

Reviewer #2: Yes

4. Have the authors made all data underlying the findings in their manuscript fully available?

Reviewer #1: Yes

Reviewer #2: Yes

5. Is the manuscript presented in an intelligible fashion and written in standard English?

Reviewer #1: Yes

Reviewer #2: No

6. Review Comments to the Author

Reviewer #1: The authors have adequately addressed my comments raised in the previous round of review. And the manuscript is technically sound, and the data support the conclusions, and the statistical analysis is performed appropriately and rigorously. So I feel that this manuscript is now acceptable for publication.

Reviewer #2: 1. Please clarify the term "plasma" in your ms. Especially the title 1.5.2.2.

2. Detail of microflora analysis should be revealed.

3. Other methods for isoflavone conversion should be touched at least.

(a) Chen, KI, Yao, Y, Chen, HJ, Lo, YC, Yu, RC and Cheng, KC. Hydrolysis of isoflavone in black soy milk using cellulose bead as enzyme immobilizer. 2016. Journal of Food and Drug Analysis 24(4):788-795.

(b) Chen, KI, Lo, YC, Liu, CW, Yu, RC, Chou, CC and Cheng, KC. Enrichment of two isoflavone aglycones in black soymilk by using coffee grounds as an immobilizer for b-glucosidase. 2013. Food Chemistry. 139:79-85.

7. PLOS authors have the option to publish the peer review history of their article (what does this mean?). If published, this will include your full peer review and any attached files.

Reviewer #1: No

Reviewer #2: No

---

## [Author Response · Author response to Decision Letter 1]

4 Jun 2021

1. If the authors have adequately addressed your comments raised in a previous round of review and you feel that this manuscript is now acceptable for publication, you may indicate that here to bypass the “Comments to the Author” section, enter your conflict of interest statement in the “Confidential to Editor” section, and submit your "Accept" recommendation.

Reviewer #1: All comments have been addressed

Reviewer #2: All comments have been addressed

Answer, Thank the reviewer for his approval of the revision of the article.

2. Is the manuscript technically sound, and do the data support the conclusions?

Reviewer #1: Yes

Reviewer #2: Yes

Answer, Thank the reviewer for his approval of the revision of the article.

3. Has the statistical analysis been performed appropriately and rigorously?

Reviewer #1: Yes

Reviewer #2: Yes

Answer, Thank the reviewer for his approval of the revision of the article.

4. Have the authors made all data underlying the findings in their manuscript fully available?

Reviewer #1: Yes

Reviewer #2: Yes

Answer, Thank the reviewer for his approval of the revision of the article.

5. Is the manuscript presented in an intelligible fashion and written in standard English?

Reviewer #1: Yes

Reviewer #2: No

Answer, Thank the reviewer for his approval of the revision of the article. The language in the article has been polished up by professionals.

6. Review Comments to the Author

Reviewer #1: The authors have adequately addressed my comments raised in the previous round of review. And the manuscript is technically sound, and the data support the conclusions, and the statistical analysis is performed appropriately and rigorously. So I feel that this manuscript is now acceptable for publication.

Reviewer #2: 1. Please clarify the term "plasma" in your ms. Especially the title 1.5.2.2.

2. Detail of microflora analysis should be revealed.

3. Other methods for isoflavone conversion should be touched at least.

(a) Chen, KI, Yao, Y, Chen, HJ, Lo, YC, Yu, RC and Cheng, KC. Hydrolysis of isoflavone in black soy milk using cellulose bead as enzyme immobilizer. 2016. Journal of Food and Drug Analysis 24(4):788-795.

(b) Chen, KI, Lo, YC, Liu, CW, Yu, RC, Chou, CC and Cheng, KC. Enrichment of two isoflavone aglycones in black soymilk by using coffee grounds as an immobilizer for b-glucosidase. 2013. Food Chemistry. 139:79-85.

Answer, The plasma here is a blood sample from a rat. This article mainly considers the conversion of isoflavones from the perspective of microorganisms.

---

## [Editor Report · Decision Letter 2]

14 Jun 2021

Effect of the HXBM408 bacteria on rat intestinal bacterial diversity and the metabolism of soybean isoflavones

PONE-D-20-06300R2

Dear Dr. Kailun,

We’re pleased to inform you that your manuscript has been judged scientifically suitable for publication and will be formally accepted for publication once it meets all outstanding technical requirements.

Kind regards,

Suzanne L. Ishaq, PhD

Academic Editor

PLOS ONE
---

## [Editor Report · Acceptance letter]

5 Jul 2021

PONE-D-20-06300R2 

Effect of the HXBM408 bacteria on rat intestinal bacterial diversity and the metabolism of soybean isoflavones 

Dear Dr. Kailun:

I'm pleased to inform you that your manuscript has been deemed suitable for publication in PLOS ONE. Congratulations! Your manuscript is now with our production department. 

Kind regards, 

on behalf of

Dr. Suzanne L. Ishaq 

Academic Editor

PLOS ONE